# END-TO-END LEARNABLE HISTOGRAM FILTERS

**Rico Jonschkowski & Oliver Brock**
Robotics and Biology Lab
Technische Universität Berlin
Berlin, Germany
{rico.jonschkowski,oliver.brock}@tu-berlin.de

## ABSTRACT

Problem-specific algorithms and generic machine learning approaches have complementary strengths and weaknesses, trading-off data efficiency and generality. To find the right balance between these, we propose to use problem-specific information encoded in algorithms together with the ability to learn details about the problem-instance from data. We demonstrate this approach in the context of state estimation in robotics, where we propose end-to-end learnable histogram filters—a differentiable implementation of histogram filters that encodes the structure of recursive state estimation using prediction and measurement update but allows the specific models to be learned end-to-end, i.e. in such a way that they optimize the performance of the filter, using either supervised or unsupervised learning.

## 1 INTRODUCTION

Traditionally, computer scientists solve problems by designing algorithms. Recently, this practice has received competition from machine learning methods that automatically extract solutions from data. One example of this development is the field of computer vision, where the state of the art is based on deep neural networks rather than on human-designed algorithms (He et al., 2015). But these two approaches to problem solving—algorithms and learning—are not mutually exclusive; in fact, they can complement each other. *Effective* problem solving exploits *all* available information, whether it be encoded in algorithms or captured by data. This paper presents a step towards tightly combining these sources of information.

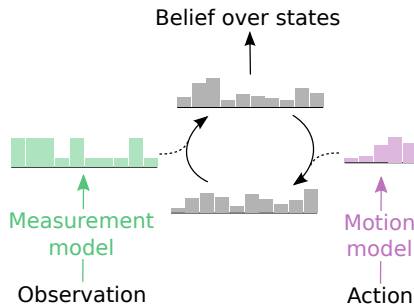

Figure 1: End-to-end learnable histogram filters. Models are learned; algorithmic structure is given.

We demonstrate the combination of problem-specific algorithms with generic machine learning in the context of state estimation in robotics. The state estimation problem exhibits a clear algorithmic structure, captured in a provably optimal way by *Bayes filters* (Thrun et al., 2005). But the use of such a filter requires the specification of a motion model and a measurement model that is specific to a particular problem instance. We want to leverage the general knowledge captured in the Bayes filter, while extracting the instance-specific models from data using deep learning (Goodfellow et al., 2016). We achieve this by implementing a differentiable version of the histogram filter—a specific type of Bayes filter that represents probability distributions with histograms—including learnable motion and measurement models (see Fig. 1). With this implementation, we can learn these models end-to-end using backpropagation, while still taking advantage of the structure encoded in Bayes filters. Interestingly, this combination also enables unsupervised learning.

Our contributions are both conceptual and technical. Our conceptual contribution is the principle of tightly combining algorithms and machine learning to balance data-efficiency and generality. Our technical contribution is the end-to-end learnable histogram filter, which enables the use of this Bayes filter variant in a more generic way. Our experiments show that our method is more data-efficient than generic neural networks, improves performance compared to standard histogram filters, and—most importantly—enables unsupervised learning of recursive state estimation loops.

## 2    COMBINING ALGORITHMS AND MACHINE LEARNING

Every information that is contained in the solution to a problem must either be provided as prior knowledge (*prior* for short) or learned from data. Different approaches balance these sources of information differently. In the classic approach to computer science, all required information is provided by a human (e.g. in the form of algorithms and models). In the machine learning approach, only a minimal amount of prior knowledge is provided (in form of a learning algorithm) while most information is extracted from data (see Fig. 2). When trading-off how much and which information should be provided as a prior or emerge from data, we should consider the entire spectrum rather than limit ourselves to these two end points.

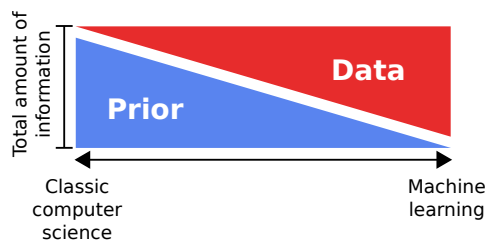

Figure 2: Information sources: prior and data

In the context of robotics, for example, it is clear that the left end of this spectrum will not enable intelligent robots, because we cannot foresee and specify every detail for solving a wide range of tasks in initially unknown environments. Robots need to collect data and learn from them. But if we go all the way to the right end of the spectrum, we need large amounts of data, which is very difficult to obtain in robotics where data collection is slow and costly. Luckily, robotic tasks include rich structure that can be used as prior. Physics, for example, governs the interaction of any robot and its environment and physics-based priors can substantially improve learning (Scholz et al., 2014; Jonschkowski & Brock, 2015). But robotic tasks include additional structure that can be exploited.

**Every algorithm that has proven successful in robotics implicitly encodes information about the structure of robotic tasks. We propose to use this robotics-specific information captured by robotic algorithms and combine it with machine learning to fill in the task-specific details based on data. By tightly combining algorithms and machine learning, we can strike the right balance between generality and data-efficiency.**

## 3    RELATED WORK

Algorithms and machine learning can be combined in different ways, using algorithms either 1) as fixed parts of solutions, 2) as parts of the learning process, or 3) as both. The first approach learns task-specific models in isolation and then combines them with algorithms in the solution. Examples for this approach are numerous, e.g. a Go player that applies a planning algorithm on learned models (Silver et al., 2016), a perception pipeline that combines the iterative closest point algorithm with learned object segmentation (Zeng et al., 2016), or robot control based on learned motion models (Nguyen-Tuong & Peters, 2011).

The second approach uses algorithms as teachers to generate training data. With this data, we can learn a function that generalizes beyond the capabilities of the original algorithm or that can be fine-tuned to a specific problem instance. For example, self-play in Go (using the algorithm as part of the solution) can be used to create new samples to learn from (Silver et al., 2016), training data for learning segmentation can be generated by simple algorithms such as background subtraction (Zeng et al., 2016), and reinforcement learning problems can be solved using training samples generated via trajectory optimization (Levine & Koltun, 2013).

The third approach—the one that we are focusing on in this paper—uses the same algorithms in the learning process and in the solution. The main idea is to optimize the models for the algorithms that use them rather than learning them in isolation. To achieve this, the algorithms need to be differentiable, such that we can compute how changes in the model affect the output of the algorithm, which allows to train the models end-to-end. This idea has been applied to different algorithms, e.g. in the form of neural Turing machines (Graves et al., 2014) and neural programmer-interpreters (Reed & de Freitas, 2015). In the context of robotics, Tamar et al. (2016) have presented a differentiable planning algorithm based on value iteration. And, most directly related to our work, Haarnoja et al. (2016) have applied this idea to Kalman filters, showing that measurement models based on visual input can be learned end-to-end as part of the filter. Our work differs from this by representing the belief with a histogram rather than a Gaussian, which allows to track multiple hypotheses—a neces-

sity for many robotic tasks. Furthermore, we focus on tasks where the robot has information about its actions and learn both the measurement model and the motion model jointly. Our paper extends an earlier workshop submission (Jonschkowski & Brock, 2016).

## 4 PRELIMINARIES: HISTOGRAM FILTERS AND OTHER BAYES FILTERS

A *Bayes filter* (Thrun et al., 2005) is an algorithm to recursively estimate a probability distribution over a latent *state* $s$ (e.g. robot pose) conditioned on the history of *observations* $o$ (e.g. camera images) and *actions* $a$ (e.g. velocity commands). This posterior over states is also called *belief*, $\text{Bel}(s_t) = p(s_t|a_{1:t-1}, o_{1:t})$. A *histogram filter* is a type of Bayes filter that represents the belief as a histogram; a discretization of the state space with one probability value per discrete state $s$. One of the key assumptions in Bayes filters is the Markov property of states, from which follows that the current belief $\text{Bel}(s_t)$ summarizes all information of the entire history of observations and actions that is relevant for predicting the future.

Other key assumptions determine how the belief is recursively updated using two alternating steps: the *prediction* step based on the last action $a_{t-1}$ and the *measurement update* step based on the current measurement $o_t$. Note that these two sources of information are separated, which results from the assumption of conditional independence of observation and action given the state.

The prediction step assumes actions to change the state according to the known motion model $p(s_t \mid s_{t-1}, a_{t-1})$. After performing an action $a_{t-1}$, the new belief for a given state $s_t$ is computed by summing over all possible ways through which state $s_t$ could have come about,

$$\overline{\text{Bel}}(s_t) = \sum_{s_{t-1}} p(s_t \mid s_{t-1}, a_{t-1})\text{Bel}(s_{t-1}). \tag{1}$$

The measurement update step assumes observations to only depend on the current state as defined by a known measurement model $p(o_t \mid s_t)$. After receiving an observation $o_t$, the belief for every state $s_t$ is updated using Bayes' rule,

$$\text{Bel}(s_t) \propto p(o_t \mid s_t)\overline{\text{Bel}}(s_t). \tag{2}$$

If motion model and measurement model are unknown, we want the robot to learn these models from data. Apart from the assumptions already mentioned, learning explicit models allow us to restrict their hypothesis space according to assumptions (e.g. linear motion). Our goal is to train these models end-to-end such that we find the models that optimize state estimation performance, while preserving the useful assumptions of Bayes filters. Towards this end, we formulate the belief, the prediction, the measurement update, and the corresponding models in the deep learning framework.

## 5 END-TO-END LEARNABLE HISTOGRAM FILTERS

An end-to-end learnable histogram filter (E2E-HF) is a differentiable implementation of a histogram filter that allows both motion model and measurement model to be learned end-to-end by backpropagation through time (Werbos, 1990). Alternatively, we can view the E2E-HF as a new recurrent neural network architecture that implements the structure of a histogram filter (see Fig. 3).

### 5.1 END-TO-END LEARNING AND DIFFERENTIABILITY

If we want to use the structure of a histogram filter as a prior and fit the measurement model and the motion model to data, we can essentially do one of two things: a) learn the models *in isolation* to optimize a quality measure of the model or b) learn the models *end-to-end*, i.e. train the models as part of the entire system and optimize the end-to-end performance.

In either way, we might want to optimize the models using gradient descent, for example by computing the gradient of the learning objective with respect to the model parameters using *backpropagation* (repeated application of the chain rule). Therefore, the motion model and the measurement model need to be differentiable regardless of whether we choose option a) or option b). For b) end-to-end learning, we need to backpropagate the gradient through the histogram filter algorithm (not to

change the algorithm but to compute how to change the models to improve the algorithm's output). Therefore, in addition to the models, the algorithm itself needs to be differentiable.

The remainder of this section describes how histogram filters can be implemented in a differentiable way and how they can be learned in isolation or end-to-end. To comply with the deep learning framework, we will define the E2E-HF using vector and matrix operations. We will also introduce additional priors for computational or data efficiency. For the sake of readability, we assume a one-dimensional state space here. All formulas can easily be adapted to higher dimensions.

## 5.2 BELIEF

The histogram over states is implemented as a vector $\boldsymbol{b}$ of probabilities with one entry per bin,

$$\boldsymbol{b}_t = [\mathrm{Bel}(S_t = 1),\ \mathrm{Bel}(S_t = 2),\ \ldots,\ \mathrm{Bel}(S_t = |S|)].$$

We can also think of the belief as a neural network layer where the activation of each unit represents the value of a histogram bin. The belief $\boldsymbol{b}_t$ constitutes the output of the histogram filter at the current step $t$ and an input at the next step $t+1$—together with an action $a_t$ and an observation $o_{t+1}$ (see Fig. 3).

## 5.3 PREDICTION (MOTION UPDATE)

The most direct implementation of the prediction step (which we replace shortly) defines a learnable function $f$ for the motion model, $f : s_t, s_{t-1}, a_{t-1} \mapsto p(s_t \mid s_{t-1}, a_{t-1})$, and employs $f$ in the prediction step (Eq. 1). The equation can be vectorized for computational efficiency by defining a $|S| \times |S|$ matrix $\boldsymbol{F}$ with $F_{i,j}(a) = f(i, j, a)$, such that $\overline{\boldsymbol{b}}_t = \boldsymbol{F}(a_{t-1})\boldsymbol{b}_{t-1}$.

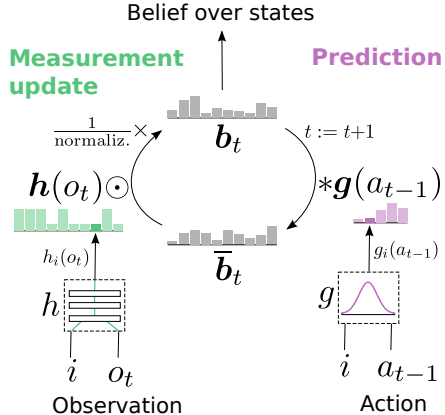

Figure 3: End-to-end learnable histogram filter. Motion model (purple) and measurement model (green) are learned; the algorithmic structure is given ($*$: convolution, $\odot$: element-wise multiplication).

However, this approach is computationally expensive because it requires $|S|^2$ evaluations of $f$ for a single prediction step. We can make this computation more efficient, if we assume robot motion to be local and consistent across the state space, i.e.

$$p(s_t \mid s_{t-1}, a_{t-1}) = p(\Delta s_t \mid a_{t-1}),$$
$$\forall t |\Delta s_t| \leq k,$$

where $\Delta s_t = s_t - s_{t-1}$ and $k$ is the maximum state change. Accordingly, we define a new learnable function for the motion model, $g : \Delta s_t, a_{t-1} \mapsto p(\Delta s_t \mid a_{t-1})$ and use $g$ instead of $f$. For vectorization, we define a $(2k+1)$-dimensional vector $\boldsymbol{g}(a)$, whose elements $g_i(a) = g(i-k-1, a)$ represent the probabilities of all positive and negative state changes up to $k$. We can now reformulate the prediction step (Eq. 1) as a convolution ($*$),

$$\overline{\boldsymbol{b}}_t = \boldsymbol{b}_{t-1} * \boldsymbol{g}(a_{t-1}),$$

where the belief $\boldsymbol{b}_{t-1}$ is convolved with the motion kernel $\boldsymbol{g}(a_{t-1})$ for action $a_{t-1}$ (see Fig. 3).

### 5.3.1 MOTION MODEL

The learnable motion model $g$ can be implemented as any feedforward network that maps $\Delta s$ and $a$ to a probability. The prior that $\boldsymbol{g}(a)$ represents a probability mass function, i.e. that the elements of $\boldsymbol{g}(a)$ should be positive and sum to one, can be enforced using the softmax nonlinearity on the vector of unnormalized network outputs $\tilde{\boldsymbol{g}}(a)$, such that $g_i(a) = \frac{e^{\tilde{g}_i(a)}}{\sum_j e^{\tilde{g}_j(a)}}$.

Another useful prior for $g$ is smoothness with respect to $\Delta s$ and $a$, i.e. that similar combinations of $\Delta s$ and $a$ lead to similar probabilities. This smoothness is the reason why (for standard feedforward networks), we should use $\Delta s$ as an input rather than as index for different output dimensions. With additional knowledge about robot motion, we can replace smoothnes by a stronger prior. For the

experiments in this paper, we assumed linear motion with zero mean Gaussian noise, and therefore defined the motion model with only two learnable parameters $\alpha$ and $\sigma$,

$$\tilde{g}(\Delta s, a) = e^{-\frac{(\Delta s - \alpha a)^2}{\sigma^2}},$$

and the obligatory normalization, $g(\Delta s, a) = \frac{\tilde{g}(\Delta s, a)}{\sum_{j=-k}^{k} \tilde{g}(j, a)}$.

## 5.4 MEASUREMENT UPDATE

Analogously to the motion model in the prediction step, we define a learnable function $h$ that represents the measurement model for the measurement update, $h : s_t, o_t \mapsto p(o_t \mid s_t)$. To vectorize the update equation (Eq. 2), we define a vector $\boldsymbol{h}(o)$ with elements $h_i(o) = h(i, o)$, such that the measurement update corresponds to element-wise multiplication ($\odot$) with this vector,

$$\tilde{\boldsymbol{b}}_t = \boldsymbol{h}(o) \odot \bar{\boldsymbol{b}}_t,$$

followed by a normalization, $\boldsymbol{b}_t = \frac{\tilde{\boldsymbol{b}}_t}{\sum_j \tilde{b}_{t,j}}$ (see Fig. 3).

### 5.4.1 MEASUREMENT MODEL

The learnable function $h$ that represents the measurement model can again be implemented by any feedforward network. Since $h$ corresponds to $p(o_t \mid s_t)$—a probability distribution over observations—it needs to be normalized across observations, not across states. To realize the correct normalization, we need to compute the unnormalized likelihood vector $\tilde{\boldsymbol{h}}(o)$ for every observation $o$ and compute the softmax over the corresponding scalars in different vectors rather than over the scalars of the same vector: $\boldsymbol{h}(o) = \frac{e^{\tilde{\boldsymbol{h}}(o)}}{\sum_{o'} e^{\tilde{\boldsymbol{h}}(o')}}$. If the observations are continuous instead of discrete, this summation must be approximated using sampled observations.

For the experiments in this paper, we represented $h$ by a network with three hidden layers of 32 rectified linear units (Nair & Hinton, 2010), followed by a linear function and a normalization as described above. Using the observation and state as input rather than output dimensions again incorporates the smoothness prior on these quantities.

## 5.5 LEARNING

We can learn the motion model $g$ and the measurement model $h$ using different learning objectives based on different sequences of data. We will first look at a number of supervised learning objectives that require $o_{1:T}$, $a_{1:T}$, $s_{1:T}$, and sometimes $x_{1:T}$—the underlying continuous state. Then, we will describe unsupervised learning that only needs $o_{1:T}$ and $a_{1:T}$.

### 5.5.1 SUPERVISED LEARNING IN ISOLATION

Both models can be learned in isolation by optimizing an objective function, e.g. the cross-entropy between experienced state change / observation and the corresponding outputs of $g$ and $h$,

$$L_g = -\frac{1}{T-1} \sum_{t=2}^{T} \boldsymbol{e}^{(\Delta s_t - k - 1)} \log(\boldsymbol{g}(a_{t-1})),$$

$$L_h = -\frac{1}{T} \sum_{t=1}^{T} \boldsymbol{e}^{(o_t)} \log(\boldsymbol{h}(o_t)),$$

where $\boldsymbol{e}^{(i)}$ denotes a standard basis vector with all zeros except for a one at position $i$, that is the position that represents the experienced state change or observation.

### 5.5.2 SUPERVISED END-TO-END LEARNING

Due to our differentiable implementation, the models can also be learned end-to-end using back-propagation through time (Werbos, 1990), which we apply on several overlapping subsequences

of length $C$ (in our experiments, $C = 32$). In the corresponding learning objectives, we compare the belief at the final time step of this subsequence with the true state. If we want to optimize the accuracy of the filter with respect to its discrete states, we can again use a cross-entropy loss,

$$L_{\text{acc.}} = -\frac{1}{T-C} \sum_{t=C+1}^{T} e^{(s_t)} \log(b_t^{(t-C:t)}),$$

where $b_t^{(t-C:t)}$ denotes the final belief at time step $t$ when the histogram filter is applied on the subsequence that spans steps $t - C$ to $t$. Alternatively, we might want to optimize other objectives, e.g. the mean square error with respect to the underlying continuous state,

$$L_{\text{mse}} = -\frac{1}{T-C} \sum_{t=C+1}^{T} (x_t - \mathbf{x} b_t^{(t-C:t)})^2,$$

where $\mathbf{x}$ denotes a vector of the continuous values to which the discrete states correspond, such that $\mathbf{x} b_t^{(t-C:t)}$ is the weighted average of these values according to the final belief in this subsequence.

### 5.5.3 Unsupervised End-To-End Learning

By exploiting the structure of the histogram filter algorithm and the differentiability, we can even train the models without any state labels by predicting future observations, but later use the models for state estimation. Similarly to supervised end-to-end learning, we apply the filter on different subsequences of length $C$, but then we follow this with $D$ steps without performing the measurement update (in our experiments, $D = 32$). Instead, we use the measurement model to predict the observations. $\text{Pred}(o_t) = \sum_{s_t} p(o_t \mid s_t) \overline{\text{Bel}}(s_t) = h(o_t) \overline{b}_t$. To predict the probabilities for all observations, we define a matrix $H$ with elements $H_{i,j} = h(i, j)$ as defined in Section 5.4. Putting everything together, we get the following loss for unsupervised end-to-end learning:

$$L_{\text{unsup.}} = -\frac{1}{(T-C)D} \sum_{t=C+1}^{T} \sum_{d=1}^{D} e^{(o_{t+d})} \log(H^\top \overline{b}_{t+d}^{(t-C:t+d)}).$$

## 6 Experiments

We consider the problem of learning to estimate the robot's state in unknown environments with partial observations. In this problem, we compare histogram filters for which the models are learned in isolation (HF), end-to-end learnable histogram filters (E2E-HFs), and two-layer long-short-term memory networks (LSTMs, Hochreiter & Schmidhuber, 1997). The models of the HFs are learned by optimizing the loss functions $L_g$ and $L_h$ presented in the previous section. For the E2E-HFs and LSTMs, we compare end-to-end learning using $L_{\text{acc.}}$, $L_{\text{mse}}$, and $L_{\text{unsup.}}$.

Our results show that 1) the algorithmic prior in HFs and E2E-HFs increases data efficiency for learning localization compared to generic LSTMs, 2) end-to-end learning improves the performance of E2E-HFs compared to HFs, and 3) E2E-HFs are able to learn state estimation without state labels.

### 6.1 Problem: Learning Recursive State Estimation in Unknown Environments

An important state estimation problem in partially observable environments is localization: a robot moves through an environment by performing actions and receives partial observations, such that it needs to filter this information over time to estimate its state, i.e. its position. In our experiments, the robot does not know the environment beforehand and thus has to learn state estimation from data.

We performed experiments in two localization tasks: a) a hallway localization task (Thrun et al., 2005) and b) a drone localization task (see Fig. 4). The tasks are similar in that they have continuous actions and binary observations (door/wall and purple/white tile), both of which are subject to $10\%$ random error. The tasks differ in their dimensionality. In the hallway task, the robot only needs to estimate a one-dimensional state (its position along the hallway), which for all methods is discretized into 100 states. The drone localization task has a two-dimensional state, which is discretized into 50 bins per dimension resulting in 2500 bins in total. The challenge in both tasks is that the door/tile

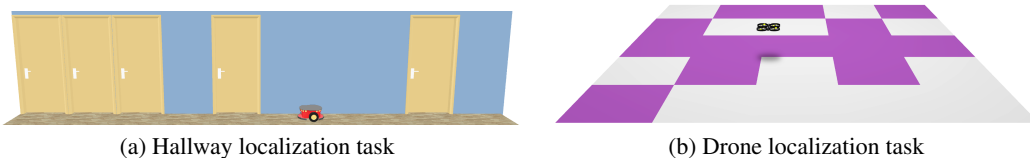

(a) Hallway localization task                    (b) Drone localization task

Figure 4: Randomly sampled environments per task. Motion and measurement models are unknown.

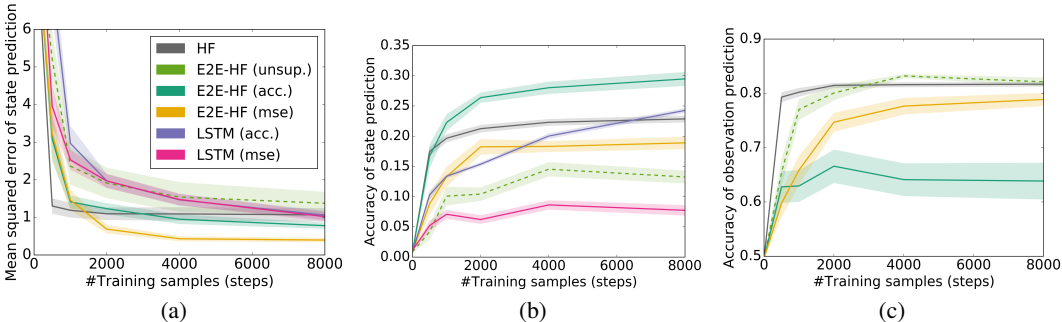

(a)                                (b)                                (c)

Figure 5: Hallway task, learning curves for different metrics: (a) mean squared error of estimating the continuous state—lower is better, (b) accuracy of estimation the discrete state—higher is better, (c) accuracy of predicting the next 32 observations—higher is better. The legend specifies both the architecture and the learning objective. Lines show means, shaded surfaces show standard errors. The dashed line highlights unsupervised learning (no state labels). LSTMs trained for state estimation cannot predict observations and therefore are not included in (c).

locations, the scale of the actions, and the amount of random noise are unknown and need to be learned from data, i.e. a sequence of observations, actions, and—in the supervised setting—states produced by the robot moving randomly through the environment. More details about the tasks, the experimental setting, learning parameters, etc. can be found in Appendix A.

## 6.2    RESULTS: IMPROVED DATA-EFFICIENCY

**Hallway task:** We performed multiple experiments in the hallway localization task with different amounts of training data. The learning curves with respect to mean squared error for supervised learning show large differences in data efficiency (see solid lines in Fig. 5a): E2E-HFs require substantially less training samples than LSTMs to achieve good performance (2000 rather than > 8000). HFs are even more data-efficient but quickly stop improving with additional data.

**Drone task:** For the drone localization task, we performed an experiment using 4000 training steps (see Table 1). Our results show that this data is sufficient for the E2E-HF (but not for the LSTM) to achieve good performance. Our method only required a similar amount of data as for the 1D hallway task, even though the histogram size had increased from 100 to 2500 bins.

**Discussion:** The priors encoded in the E2E-HF improve data efficiency because any information contained in these priors does not need to be extracted from data. This leads to better generalization, e.g. the ability to robustly and accurately track multiple hypotheses (see Fig.6).

**Note on computational limits:** The size of the histogram is exponential in the number of state dimensions. A comparison between the 1D and the 2D task suggests that data might not be the bottleneck for applying the method to higher dimensional problems, since the data requirements were similar. However, the increased histogram size directly translates into longer training times, such that computation quickly becomes the bottleneck for scaling this method to higher-dimensional problems. Addressing this problem will require to change the belief representation, e.g. to particles or a mixture of Gaussians, which is an important direction for future work.

| Method | MSE (state) | Acc. (state) | Acc. (obs.) |
|---|---|---|---|
| HF | 0.22 | 0.05 | **0.81** |
| E2E-HF (unsup.) | 0.22 | 0.03 | **0.81** |
| E2E-HF (acc.) | 0.39 | **0.17** | 0.40 |
| E2E-HF (mse) | **0.16** | 0.08 | 0.66 |
| LSTM (acc.) | 3.03 | 0.03 | – |
| LSTM (mse) | 0.50 | 0.06 | – |

Table 1: Drone task: test performance of different methods with 4000 training samples

## 6.3 RESULTS: OPTIMIZATION OF END-TO-END PERFORMANCE

**Hallway task:** While HFs excel with very few data, E2E-HFs surpass them if more than 2000 training samples are available (see gray and yellow lines in Fig. 5a). For the mean squared error metric, the best method is the E2E-HF with a mean squared error objective (yellow line). However, if we care about a different metric, e.g. accuracy of estimating the discrete state, the methods rank differently (see Fig. 5b). The best method for the previous metric (yellow line) is outperformed by HFs (gray line) and even more so by E2E-HFs that are optimized for accuracy (teal line). For yet another metric, i.e. accuracy of predicting future observations, HFs outperform both other approaches but are equal to E2E-HFs optimized for predicting future observations (see Fig. 5c).

**Drone task:** The results of the drone localization task show the same pattern (see Table 1). The best method for every metric is the E2E-HF that optimizes this metric.

**Discussion:** E2E-HFs perform better than HFs because they optimize the models for the filtering process (with respect to the metric they were trained for) rather than optimizing model accuracy. This can be advantageous because "inaccurate" models can improve end-to-end performance (compare the HF model learned in isolation to the models learned end-to-end in Fig. 6a).

## 6.4 RESULTS: ENABLING UNSUPERVISED LEARNING

**Hallway and drone tasks:** In both tasks, unsupervised E2E-HFs were similar to HFs and better than all other methods for predicting future observations. Interestingly, they also had comparatively low mean squared error for state estimation even though they had never seen any state labels (see dashed green line in Fig. 5 and second line in Table 1). In fact, the qualitative results for both tasks show a remarkable similarity between the learned models and the estimated belief between HFs and unsupervised E2E-HFs (compare HF and E2E-HF (unsup.) in Fig. 6) and Fig. 7.

**Discussion:** E2E-HFs can learn state estimation purely based on observations and actions. By predicting future observations using the structure of the histogram filter algorithm, the method discovers a state representation that works well with this algorithm, which is surprisingly close to the "correct" models learned by HFs, although no state labels are used.

## 7 CONCLUSION

We proposed to tightly combine prior knowledge captured in algorithms with the ability to learn from data. We demonstrated the feasibility and the advantages of this idea in the context of state estimation in robotics. Algorithmic priors lead to data-efficient learning, as knowledge about the problem structure encoded in the algorithm is provided explicitly and does not have to be extracted from data. The ability to learn from data enables the use of algorithms when task-specifics are unknown. The tight combination of both improves performance as the models are optimized for use in the algorithm. Furthermore, the explicit algorithmic structure enables unsupervised learning. We view our results as a proof of concept and are convinced that the combination of algorithms and machine learning will help solve novel problems, while balancing data efficiency and generality.

### ACKNOWLEDGMENTS

We gratefully acknowledge the funding provided by the Alexander von Humboldt foundation and the Federal Ministry of Education and Research (BMBF).

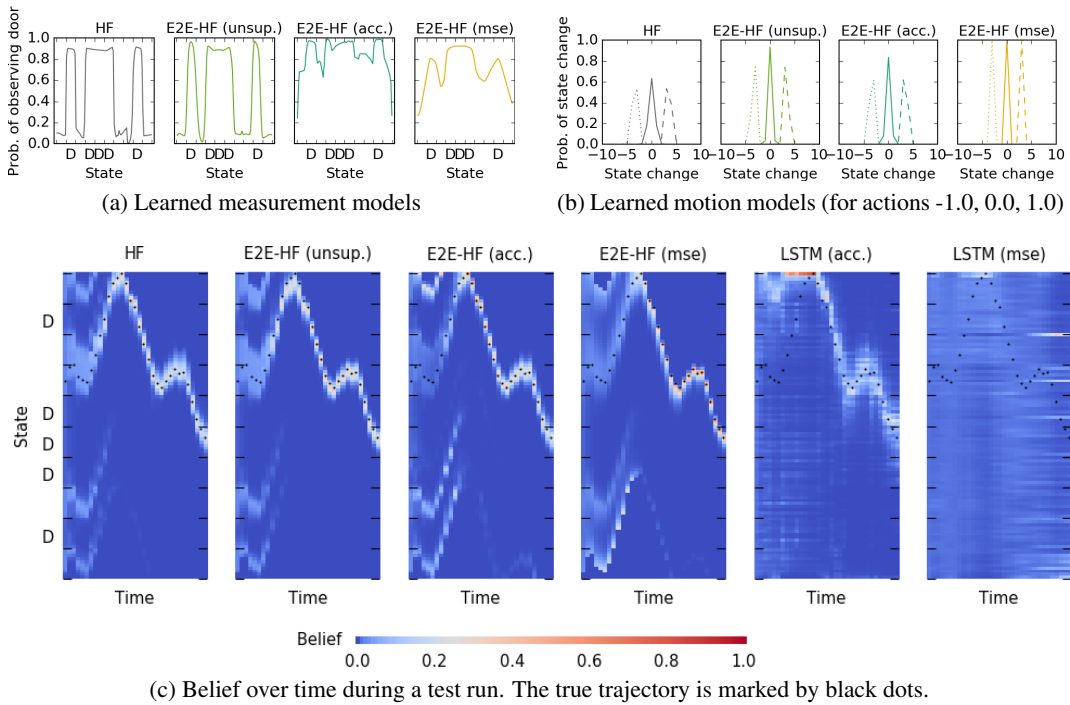

(a) Learned measurement models
(b) Learned motion models (for actions -1.0, 0.0, 1.0)

(c) Belief over time during a test run. The true trajectory is marked by black dots.

Figure 6: Hallway navigation task: (a-b) learned models for one environment (D=door state) and (c) belief evolution for a single test run in this environment. All methods used 4000 training samples.

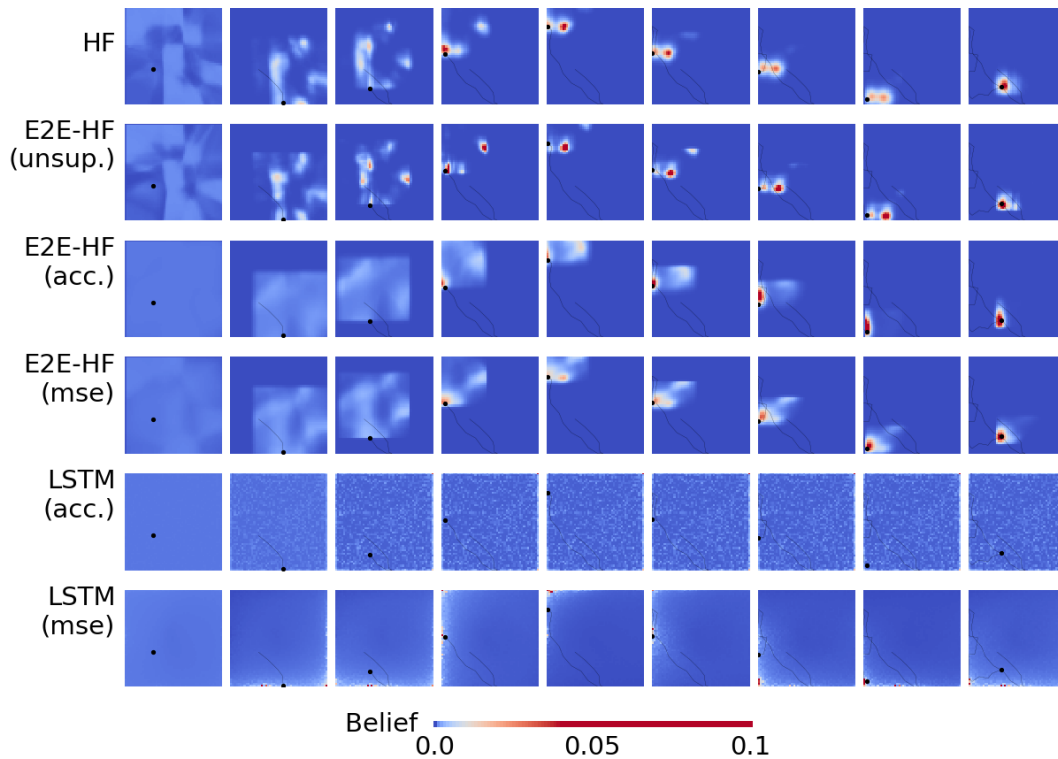

Figure 7: Drone localization task: belief evolution during single test run for different methods. Black dots/lines show the true position/trajectory of the drone. All methods used 4000 training samples.

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

## A   ADDITIONAL EXPERIMENT DETAILS

### A.1   HALLWAY LOCALIZATION TASK

The hallway has a length of 10 meters, where every full meter is either occupied by a door or by a wall. At the beginning of every experiment trial, 5 doors are randomly arranged in the 10 spots in the hallway. The binary observation of the robot senses whether the center of the robot is next to a door or next to a wall. With probability 0.1, the observation returns the wrong information, e.g. "wall" instead of "door" if the robot is next to a door.

The robot is represented as a single point. It moves with a velocity between -1 and 1 meter per time step and stops when it reaches either end of the hallway. The action information that the robot receives is the step that it performed as measured by odometry. This odometry measurement is corrupted with zero mean Gaussian noise with standard deviation of 10% of it's actual movement. Additionally, the odometry is scaled by a number between 0.5 and 5.0, which is randomly sampled at the beginning of every trial, i.e. the robot does not know its exact embodiment. This makes the exact motion model unknown, such that the robot needs to learn it from data.

Both during training and during testing the robot moves randomly, i.e. it randomly accelerates by a value between -0.5 and 0.5 at each time step. Apart from this acceleration, its velocity is affected by 10% friction at each time step and is set to zero when the robot reaches either end of the hallway. For each trial, the training data consists of a single random walk of the robot of length between 500 steps and 8000 steps. The data for the unsupervised learning, includes only the sequence of noisy observations and actions. For supervised learning, it additionally includes the groundtruth continuous and discrete state, i.e. the position of the robot.

The test data consisted of 1000 short time sequences of the robot moving in the same fashion starting from a random position. For all performance metrics, the belief was tracked for 32 steps. For the metric that measured observation prediction accuracy, the task was to predict 32 future observations given a sequence of 32 actions based on the current belief.

### A.2   DRONE LOCALIZATION TASK

The area for the drone localization task has a size of 5 times 5 meters, where every one meter tile is either purple or white. At the beginning of every experiment, the color of each tile is decided by a fair coin flip. Analogously to the hallway task, the binary observations inform the robot about the color of the tile which is directly underneath it. With probability of 0.1, this observation returns the wrong color.

The drone is represented as a single point in 2D space. It moves with velocities between -0.5 and 0.5 meter per time step and stops when it reaches the boundary of the area. The other aspects of its movement, the noisy odometry, and the movement generation for training and test data are analogous to the hallway localization task.

### A.3   EXPERIMENTAL DETAILS

**LSTM baseline:** The LSTM baseline consists of two LSTM layers with 32 units per layer, followed by a softmax layer.

**Training procedure:** All methods where trained via minibatch stochastic gradient descent with batch size 32 using Adam (Kingma & Ba, 2014) with learning rate 0.001. The training length was determined using early stopping with patience, where 20% of the training data was used for validation. After 100 epochs without an improvement on the validation data, the parameters that achieved highest validation performance were returned.

### A.4   SOFTWARE

We used v-rep for simulation (E. Rohmer, 2013) and theano (Theano Development Team, 2016) with Lasagne (Dieleman et al., 2015) as deep learning framework for our implementation.

