# Peer review of "End-to-End Learnable Histogram Filters"

_ICLR 2017 — rejected_

[Official Review · AnonReviewer3 · rating 3 · confidence 3 · 15 Dec 2016]
**No Title**

The authors propose a time-series model with discrete states for robotics applications. I think the proposed method is too simplistic to be useful in the presented form, eg. 1) the state space (dimensionality & topology) is exactly matched to the experiments 2) displacements in the transition model are linear in the actions 3) observations are one-dimensional. This seems to be quite behind the current state of the art, eg “Embed to Control” by Watter et al 2015, where a state representation is learned directly from pixels.
Furthermore the authors do not compare to any other method except for an out-of-the-box LSTM model. Also, I feel like there must be a lot of prior work for combining HMMs + NNs out there, I think it would be necessary for the authors to relate their work to this literature.

[Official Review · AnonReviewer1 · rating 4 · confidence 3 · 16 Dec 2016]

Summary:
--------
The authors propose a histogram based state representation with differentiable motion models and observation updates for state tracking from observations. Linear model with Gaussian noise is used as the motion model, while a neural network is used to learn the measurement model. They track robot states in: (1) 1-D hallway, and (2) a 2D arena.


Positives:
----------
1. Show how to encode prior knowledge about state-transitions in the architecture.
2. No assumptions about the observation model, which is learned purely from data.
3. Better accuracy than baselines with limited training data.

Negatives:
----------
1. The motion model is too simplistic. The authors in their response to earlier questions say that a generic feed-forward neural network could be used to model more complicated motions. However, then the novelty of their framework is not clear -- as then the proposed model would just be a couple of neural networks to learn the motion and observation models.

2. The observation model again is too simplistic (e.g., one dimensional observations), and is proposed to be a generic feed-forward network. Here again, the technical novelty is not clear.

3. The histogram based representation is not scalable as also highlighted by the authors. Hence, the proposed approach as it is, cannot be applied to more complicated settings.

4. In Figure 5(a,b), where they compare the state-estimation accuracy with other baselines (i.e., LSTMs), it is clear that the accuracy of the LSTM has not saturated, while that of their model has. They should do larger scale experiments with more training data (e.g., 10k,100k,500k samples). 
Note that while sample efficiency is a desirable property (also discussed in Section 6.2), we do expect models with prior knowledge to work better for small number of samples than models which do not assume any structure. Experiments with larger number of samples would be insightful.

[Official Review · AnonReviewer2 · rating 4 · confidence 3 · 30 Dec 2016]
**No Title**

Summary: This paper presents a differentiable histogram filter for state estimation/tracking. The proposed histogram filter is a particular Bayesian filter that represents the discretized states using beliefs. The prediction step is parameterized by a locally linear and translation-invariant motion model while the measurement model is represented by a multi-layered neural network. The whole system is learned with both supervised and unsupervised objectives and experiments are carried out on two synthetic robot localization tasks (1D and 2D). The major claim of this paper is that the problem-specific model structure (Bayesian filter for state estimation) should improve pure deep learning approach in data-efficiency and generalization ability. 
+This paper has nice arguments about the importance of prior knowledge to deep learning approach for specific tasks. 
+An end-to-end histogram filter is derived for state estimation and unsupervised learning is possible in this model.
-This paper seems to have a hidden assumption that deep learning (RNN) is a natural choice for recursive state estimation and the rest of paper is built upon this assumption including LSTM baselines. However, this assumption itself may not be true, because Bayesian filter is a first-established approach for this classic problem, so it it more important to justify if deep learning is even necessary for solving the tasks presented. This requests pure Bayesian filter baselines in the experiments. 
-The derived histogram filter seems to be particularly designed for discretized state space. It is not clear how well it can be generalized to continuous state space using the notation "x". More interestingly, the observation is discrete (binary) as well, which eventually makes it possible to derive a closed-form measurement update model. This setup might be too constrained. Generalizing to continuous observations is not a trivial task, not even to mention using images as observations like Haarnoja et al 2016. These design choices overall narrow down the scope of applicability.

[Author Response · Rico Jonschkowski · 30 Jan 2017]
**Retraction**

We are retracting our paper "End-to-End Learnable Histogram Filters" from ICLR to submit a revised version to another venue.

[Final Decision · Program Chairs · 06 Feb 2017]
**ICLR committee final decision**

In many respects, this is a strong paper, in my opinion better than the reviews thus far in the system suggest. The idea of learning the parameters of a state estimation system, even if it is a simple example like a histogram filter, is an interesting idea from both the ML and robotics perspectives.
 
 However, I think it's also fairly clear (from the reviews if nothing else), that substantial additional work needs to be done if this paper is to convey these ideas clearly and impactfully to the ICLR community. The value of the histogram filter may be obvious to the robotics community (though even there it seems like it would be worth pursuing the extension of these ideas to the case e.g. of particle filters), as these have historically been an important conceptual milestone in robotic perception, but the examples shown in this paper are extremely simplistic. Given that histogram filters inherently scale poorly with dimension, it's not clear from this paper itself why the techniques show particular promise for scaling up to realistic domains in the future.
 
 Pros:
 + Interesting idea of using training a state estimator end-to-end for robotics tasks
 
 Cons:
 - Histogram filters, while well-motivated historically from the robotics standpoint, are really toy examples at this point except in a very small number of settings.
 - The results aren't all that compelling, showing modest improvement (over seemingly not-very-tuned alternatively approaches), on fairly simple domains.